# Sparse Cocktail: Every Sparse Pattern Every Sparse Ratio All At Once

## Abstract

Sparse Neural Networks (SNNs) have received voluminous attention for mitigating the explosion in computational costs and memory footprints of modern deep neural networks. Despite their popularity, most state-of-the-art training approaches seek to find a single high-quality sparse subnetwork with a preset sparsity pattern and ratio, making them inadequate to satiate platform and resource variability. Recently proposed approaches attempt to jointly train multiple subnetworks (we term as "sparse co-training") with a fixed sparsity pattern, to allow switching sparsity ratios subject to resource requirements. In this work, we take one more step forward and expand the scope of sparse co-training to cover diverse sparsity patterns and multiple sparsity ratios *at once*. We introduce **Sparse Cocktail**, the first sparse co-training framework that co-trains a suite of sparsity patterns simultaneously, loaded with multiple sparsity ratios which facilitate harmonious switch across various sparsity patterns and ratios at inference depending on the hardware availability. More specifically, Sparse Cocktail alternatively trains subnetworks generated from different sparsity patterns with a gradual increase in sparsity ratios across patterns and relies on an *unified mask generation process* and the *Dense Pivot Co-training* to ensure the subnetworks of different patterns orchestrate their shared parameters without canceling each other's performance. Experiment results on image classification, object detection, and instance segmentation illustrate the favorable effectiveness and flexibility of Sparse Cocktail, pointing to a promising direction for sparse co-training. Codes will be released.

## 1 Introduction

Deep neural networks are boosted by the ever-larger model size (Brown et al., 2020; Ramesh et al., 2022; Du et al., 2022; Jumper et al., 2021). Despite their impressive performance, these gigantic models require prohibitive costs to train and infer, pushing the model size beyond the reach of common hardware. Sparsity serves as a leading concept to shrink model sizes with a negligible performance drop. By pruning a large fraction of parameters from a well-trained neural network, the resulting sparse neural networks enjoy significant computational and memory reduction at inference (Mozer & Smolensky, 1989; Han et al., 2015; Molchanov et al., 2016). Recently, as the financial and environmental costs of model training grow exponentially (Strubell et al., 2019; Patterson et al., 2021), people start to pursue training efficiency by inducing sparsity during the early training phase (Gale et al., 2019; You et al., 2019; Liu et al., 2021b) or before training (Mocanu et al., 2018; Evci et al., 2020; Lee et al., 2018; Tanaka et al., 2020). These sparse training approaches launch the new pursuit of end-to-end saving potential for both training and inference stages.

While many sparse training methods aim to pinpoint a singular optimal sparse subnetwork for a specific sparsity **pattern** (*e.g.*, unstructured, semi-structured, or structured) and **ratio** (*i.e.*, the percentage of zero elements), they often fall short in accommodating the diverse platform and resource constraints encountered in the real-world deployment of sparse neural networks. Relying on post-training pruning tailored to each hardware or constraint can be impractical, as it necessitates a unique pruning strategy for every scenario and many resource-limited platforms cannot sustain multiple model alternatives. In response, emerging research offers methods that extract multiple sparse subnetworks from a single training cycle (Chen et al., 2021; Peste et al., 2021; Miao et al., 2021; Yang et al., 2022; Dao et al., 2022). These resulting dense or sparse subnetworks can be swiftly toggled per inference requirements. We refer to those methods as **sparse co-training** for simplicity.

Early sparse co-training efforts (Yu et al., 2018; Yu & Huang, 2019; Yang et al., 2021) embed smaller subnetworks (with higher channel-level sparsity) within larger ones (with lower channel-level sparsity). This joint training with selective switching yields a set of channel-sparse networks at varying ratios in addition to the dense variant. AC/DC (Peste et al., 2021) pairs and co-trains a dense network with a pre-determined sparse subnetwork through group partitioning, alternating between compression and decompression. Conversely, AST (Yang et al., 2022) utilizes the prune-and-regrow mechanism (Liu et al., 2021c) to co-train an array of masks of different sparsities, ensuring gradient alignment between them. Both AC/DC (Peste et al., 2021) and AST (Yang et al., 2022) initially showcased their methods for unstructured sparsity before adapting them to $N{:}M$ structured sparsity. OTO (Chen et al., 2021) can prune a trained network to any channel-level sparsity ratio in a single attempt, eliminating the need for re-training. (Miao et al., 2021) achieved a similar outcome but centered on unstructured sparsity. Recently, Monarch (Dao et al., 2022) employed a hardware-efficient parameterization of dense weight matrices, specifically using the multiplication of two block-diagonal matrices, generating both dense and hardware-optimized sparse models in a single pass.

Despite advancements, current sparse co-training methodologies are fragmented. Most are confined to **one** sparsity pattern per run, and only a handful can yield multiple sparsity ratios alongside the dense version. *We contend that* there's a pressing need to broaden the scope of existing sparse co-training techniques to simultaneously encompass a wider variety of sparsity patterns and ratios. This belief stems from several factors. Firstly, real-world hardware resources can fluctuate immensely based on the specifics of an application. Secondly, sparse accelerators differ in design, each optimized for distinct sparsity patterns, such as unstructured sparsity (Liu et al., 2021c), group-wise (Rumi et al., 2020), channel-wise (Li et al., 2016), and $N{:}M$ sparsity (Nvidia, 2020). For instance, while unstructured sparsity shows promising acceleration on CPUs (DeepSparse, 2021; Liu et al., 2021c), its GPU support is considerably thinner, especially when juxtaposed against structured sparsity. Lastly, the resource needs and provisions of an ML system evolve over time, necessitating the ability for "in-situ" adaptive toggling between different sparsity ratios to meet dynamic system demands.

We hereby present **Sparse Cocktail**, a sparse co-training framework that is capable of concurrently producing multiple sparse subnetworks across a spectrum of sparsity patterns and ratios, in addition to the dense model. Our approach alternates between various sparsity pattern training phases, meanwhile incrementally raising the sparsity ratio across these phases. Underlying the multi-phase training is a unified mask generation process that allows seamless phase transitions without performance breakdown. This is complemented by a dense pivot co-training strategy augmented with dynamic distillation, aligning the optimization trajectories of diverse sparse subnetworks. In the end, all sparse subnetworks share weights from the dense network. This culminates in a "cocktail" of dense and sparse models, offering a highly storage-efficient ensemble. Our primary contributions are as follows:

- We introduce **Sparse Cocktail**, a novel sparse co-training approach that produces a diverse set of sparse subnetworks with various sparsity patterns and ratios at once. Different from previous sparse (co-)training approaches which only focus on one, at most two, types of sparsity patterns, and/or with different sparsity ratios, Sparse Cocktail co-trains a suite of sparsity patterns simultaneously, and each coming at a series of sparsity ratios. One can handily choose the desired sparsity pattern and ratio at inference based on the target hardware type and resource availability.

- Sparse Cocktail alternatively trains subnetworks generated from different sparsity patterns , meanwhile gradually increasing the sparsity ratios for all. We use a unified mask generation method and a dense pivot co-training scheme with dynamic distillation to ensure the subnetworks of different patterns and ratios orchestrate their shared parameters so that they will not cancel each other's performance. Within each sparsity pattern, we additionally perform selective weight interpolation of multiple subnetworks across different sparsity ratios, to strengthen performance further.

- Our new framework, besides essentially generalizing and "encapsulating" previous sparse co-training methods, achieves great parameter efficiency and comparable Pareto-optimal trade-off individually achieved by those methods too. For example, for co-training at different sparsity ratios, Sparse Cocktail is on par with or even outperforms strong baselines such as AST (Yang et al., 2022) and MutualNet (Yang et al., 2021). In contrast with methods that only co-train a dense/sparse network pair, Sparse Cocktail also achieves competitive performance.

## 2 RELATED WORK

### 2.1 OVERVIEW OF SPARSE TRAINING

**Dense-to-Sparse Training.**  Dense-to-sparse training begins with a dense model and progressively sparsifies it throughout the training process. Gradual magnitude pruning (GMP)(Zhu & Gupta, 2017; Gale et al., 2019) incrementally sparsifies the neural network to achieve the target sparsity over the training duration. Techniques leveraging $\ell_0$ and $\ell_1$ regularization to penalize parameters diverging from zero have also been effective in yielding compact yet performant sparse neural networks (Louizos et al., 2018; Wen et al., 2016). During training, trainable masks can be learned (Srinivas et al., 2017; Liu et al., 2020; Savarese et al., 2019; Xiao et al., 2019) and, intriguingly, even at initialization (Ramanujan et al., 2020; Chijiwa et al., 2021; Huang et al., 2022) to produce the desired SNNs.

The lottery ticket hypothesis (LTH)(Frankle & Carbin, 2018) can be broadly classified under dense-to-sparse training. LTH employs Iterative Magnitude Pruning (IMP)(Han et al., 2015) combined with weight rewinding to accurately identify high-quality sparse subnetworks (often referred to as winning tickets). When trained in isolation, these subnetworks can match the performance of the dense neural network. Techniques such as Lottery Pools (Yin et al., 2022) have further shown that most LTH solutions (i.e., converged subnetworks) reside within the same local basin. Consequently, they can be selectively interpolated to enhance LTH's performance. More recently, (Chen et al., 2022) introduced two post-training operations: weight refilling and weight regrouping. These effectively transition the benefits of unstructured sparsity to GPU-compatible sparsity patterns.

**Sparse-to-Sparse Training.**  Sparse-to-sparse training, in contrast, begins with and maintains a sparse neural network throughout training, aiming for potential end-to-end efficiencies during both training and inference. Dynamic Sparse Training (DST) (Mocanu et al., 2018; Liu et al., 2021c) has emerged as a promising strategy to derive high-performing sparse networks without the need for any dense pre-training or fine-tuning phases. Most DST techniques employ a prune-and-regrow operation(Mocanu et al., 2018) to enhance the efficacy of sparse masks. SNFS (Dettmers & Zettlemoyer, 2019) and RigL (Evci et al., 2020) notably augment the performance of DST by utilizing gradient data to cultivate weights. ITOP (Liu et al., 2021d) underscores the essential role of parameter exploration in sparse training, emphasizing that the performance of sparse training is intrinsically tied to the total number of parameters it engages with during training. Top-KAST (Jayakumar et al., 2020) exclusively updates a minor fraction of gradients during backpropagation, bypassing the need to compute dense gradients. A review of various existing sparsity patterns is in Appendix A.

### 2.2 SPARSE CO-TRAINING: MORE THAN ONE SPARSITY PATTERNS OR RATIOS AT ONCE

Existing sparse co-training methods can be divided into two paradigms: (i) co-training dense and sparse networks, and (ii) co-training multiple sparse networks from scratch.

The first paradigm encompasses methods such as S-Net (Yu et al., 2018), US-Net (Yu & Huang, 2019), and MutualNet (Yang et al., 2021). In these methods, smaller subnetworks are nested within larger ones and are co-trained through selective switching or random sampling. Partial-SGD (Mohtashami et al., 2022) employs a mix of parameter perturbation and gradient masking to co-train a full-rank dense model alongside a low-rank sparse model. In contrast, AC/DC (Peste et al., 2021) co-trains a dense network and its subnetwork with a predefined sparsity, utilizing group partitioning and alternating compression/decompression techniques.

The second paradigm, which involves co-training multiple sparse networks from scratch, features methods such as AST (Yang et al., 2022). AST employs a prune-and-regrow mechanism, enabling the co-training of several sparse subnetworks with gradient alignment between consecutive mini-batches. Monarch (Dao et al., 2022) deploys dense matrix approximation with permutable block-diagonal sparse matrices, obtaining both dense and numerous sparse models simultaneously. Cosub (Touvron et al., 2022) suggests training two random subsets of all network layers with mutual distillations in each mini-batch, yielding depth-wise sparse models and a more potent dense model.

However, several issues prevail in current sparse co-training methods: (1) the limited number of co-trainable subnetworks due to simplistic alternative or joint training, and (2) their focus on a single sparsity pattern during one training pass. These issues render them unsuitable for generating more sparse subnetworks that can cater to the requirements of diverse hardware platforms.

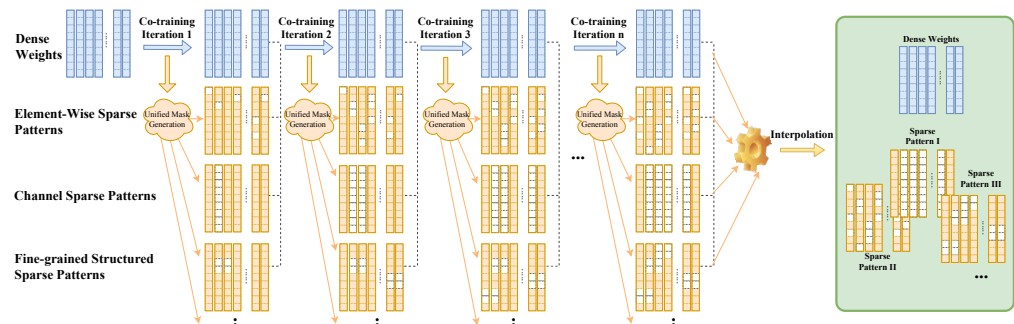

Figure 1: The flow diagram of Sparse Cocktail. Before each co-training iteration, we first perform iterative pruning with weight rewinding and the **Unified Mask Generation** technique. This produces a set of sparse subnetworks with various sparsity patterns, gradually increasing in sparsity ratios. During each co-training iteration, we use **Dense Pivot Co-training** to train subnetworks with different sparsity patterns alternatively, using a dense network as a pivot. Once all co-training steps are complete, we introduce Greedy Subnetwork Interpolation to boost the performance of the subnetworks. The final output of Sparse Cocktail is formed by a set of dense neural networks, each accompanied by multiple sparse masks with various patterns.

Besides the aforementioned, we would like to mention other loosely related research areas: $(i)$ training a pruning-friendly network for one-shot pruning without re-training (Chen et al., 2021; Miao et al., 2021), $(ii)$ neural architecture search (NAS)-based pruning that produces a multitude of subnetworks with shared parameters, albeit at a significantly higher training cost, such as OFA (Cai et al., 2019) and BigNAS (Yu et al., 2020), $(iii)$ leveraging dedicated structured pruning and distillation for iterative pruning of networks without re-training (Kurtic et al., 2023), and $(iv)$ input resolution-switchable networks, such as RS-Net (Wang et al., 2020) and again, MutualNet (Yang et al., 2021).

# 3 METHODOLOGY

## 3.1 PRELIMINARIES AND NOTATIONS

**Notations.** Let $D$ and $S$ denote the dense and sparse networks, respectively. Operations commonly referenced in pruning literature, namely pruning, weight rewinding, and re-training, are represented by $\mathcal{P}$, $\mathcal{R}$, and $\mathcal{T}$, which we'll elaborate on subsequently. To differentiate between iterations of pruning and re-training, $D$ and $S$ can be subscripted with $k = 1, 2, ..., N$. Various sparsity patterns are symbolized as $S^u$, $S^c$, and $S^{nm}$, standing for unstructured sparsity, channel-wise sparsity, and $N$:$M$ sparsity, respectively. With $m$ representing the binary masks of each sparse network and $\odot$ signifying the unstructured product operation, the sparse neural network at the $k^{th}$ iteration can be expressed as $S_k = D_k \odot m_k$.

**Iterative Magnitude Pruning.** Iterative magnitude pruning (IMP) (Han et al., 2015) iteratively prunes a dense network $D_0$ using a ratio $p$ (*e.g.*, 20%), yielding a sequence of **nested** masks with progressively increasing sparsity ratios. After each pruning step, retraining the sparse subnetwork is typically essential to restore performance. IMP received renewed attention through the Lottery Ticket Hypothesis (LTH) (Frankle & Carbin, 2018). LTH reveals that sparse subnetworks derived from IMP can achieve the performance of the dense network when trained independently with their original initializations. The power of LTH was further enhanced by weight and learning rate rewinding (Frankle et al., 2020; Renda et al., 2020). Formally, subnetworks produced by IMP through rewinding can be defined as $S_k | S_k = \mathcal{T}_k(\mathcal{R}(\mathcal{P}k(Sk-1))), S_0 = D_0, k = 1, 2, ..., N$. In this work, we extend our iterative pruning scope to craft not just one specific mask type but multiple sparsity patterns concurrently. Thus, it yields four distinct network series: $D_k, S_k^u, S_k^c, S_k^{nm}$ for $k = 0, 1, ..., N$. These represent the dense network and the sparse networks with unstructured sparsity, channel-wise sparsity, and $N$:$M$ sparsity, respectively.

## 3.2 OVERVIEW OF SPARSE COCKTAIL

The workflow of Sparse Cocktail is depicted in Fig. 1, comprising three main modules:

① **Prior to Iterative Co-training:** Before embarking on each iterative co-training phase, Sparse Cocktail first initializes three subnetworks, each embodying a unique sparsity pattern: $S_0^u, S_0^c, S_0^{nm}$, all stemming from a pre-trained dense network $D_0$ via IMP. Magnitude pruning gives rise to the unstructured $S_k^u$ and $N{:}M$ sparse $S_k^{nm}$. The channel-wise subnetworks are found by transforming $S_k^u$ and $S_k^{nm}$ through the Unified Mask Generation (UMG) process, detailed in subsequent sections.

② **During Iterative Co-training:** Within each iterative co-training phase, we rewind the weights, typically to around the $5^{th}$ epoch, and alternately train subnetworks of diverse patterns. To ensure the co-training remains stable, we intersperse with a one-step update of a dense neural network. This "dense pivot" acts as a lubricant, streamlining the sparse co-training for enhanced efficacy.

③ **Post Iterative Co-training:** Upon concluding the comprehensive iterative co-training regimen, we employ an interpolation-driven network merging technique to further augment the performance of the resultant (sub-)networks.

We shall clarify that the focus of Sparse Cocktail is **NOT** training efficiency, but instead, the ability to generate multiple network options at once - for the goal of adaptive deployment and efficient inference, same as prior works Yang et al. (2021); Peste et al. (2021); Yang et al. (2022).

## 3.3 Using Iterative Pruning with Weight Rewinding for Sparse Co-training

One challenge of training multiple diverse sparse subnetworks resembles multi-task learning: when a single parameter exists in multiple subnetworks simultaneously, it can induce conflicting gradient directions, a phenomenon observed by Yang et al. (2022). The challenge is amplified as we augment more co-trained subnetworks, especially given the blend of sparsity patterns and ratios.

To circumvent the gradient conflicts, we embrace iterative pruning, veering away from the one-shot gradual pruning methods (Yang et al., 2022; Liu et al., 2021b). This strategy ensures that subnetworks of different sparsity ratios are segregated and nurtured across discrete iterations. Moreover, with weight rewinding, we ensure these subnetworks to originate from a **unified starting point**, harmonizing their optimization trajectories and diminishing the chances of training discord.

However, our approach goes beyond producing multiple sparsity ratios; it also grapples with the tandem training of assorted sparsity patterns. Guaranteeing that these patterns are cultivated and honed without adversely affecting each other's performance is crucial. In pursuit of this, we unveil three more cornerstone techniques: **Unified Mask Generation**, **Dense Pivot Co-training**, and **Sparse Network Interpolation** - those will be detailed next.

## 3.4 Unified Mask Generation

A key question that emerges prior to iterative co-training is the methodological generation of masks with disparate sparsity patterns $m_k^u, m_k^c, m_k^{nm}$ in a way not adversely influencing one another. Pursuing independent generation for each might lead to divergent optimization trajectories for $S_k^u, S_k^c, S_k^{nm}$. In response to this challenge, we introduce the Unified Mask Generation (UMG) mechanism, designed to jointly produce $m_k^u, m_k^c, m_k^{nm}$ grounded on the criterion of individual weight magnitudes:

For the unstructured and $N{:}M$ patterns, the masks $m_k^u$ and $m_k^{nm}$ are crafted by selecting individual weights based on their magnitudes. It's worth noting that weight magnitudes are globally ranked for unstructured sparsity. In contrast, for the $N{:}M$ pattern, magnitudes are locally ranked across every contiguous set of $M$ weight elements.

The channel-wise mask $m_k^c$ presents a unique nuance: the channels to prune cannot be pinpointed based on individual weights alone. To address this, we lean on the *weight refilling* approach (Chen et al., 2022). Here, the non-pruned weights of both unstructured and $N{:}M$ patterns guide the decision on which channels to eliminate. Explicitly, for a channel $C \in \mathbb{R}^{i \times h \times w}$ (with $i$ denoting the number of input channels and $h, w$ representing the dimensions of weight kernels), the channel's importance is gauged by $\beta ||m^u \odot C||_1 + (1 - \beta)||m^{nm} \odot C||_1$. Here, $m^u$ and $m^{nm}$ are the respective unstructured and $N{:}M$ masks of this channel, with the empirical value of $\beta$ set at 0.8. Consequently, for each layer, a subset of channels showcasing peak scores—based on the predetermined channel-wise sparsity ratio—is chosen. This selection informs the composition of the channel-wise mask $m_k^c$.

Our analysis in Fig. 5 demonstrates that UMG substantially reduces the optimization conflicts between different sparsity patterns.

## 3.5 DENSE PIVOT CO-TRAINING

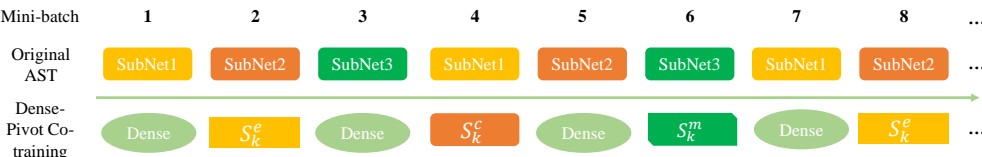

Figure 2: The comparison between AST (Yang et al., 2022) and dense pivot co-training. AST switches among subnetworks of different sparsity levels yet within the same sparsity pattern cyclically (it does not consider multiple sparse patterns). Dense pivot co-training inserts dense training steps between switching different sparse patterns.

In (Yang et al., 2022), it was highlighted how SGD training imparts an implicit regularization of gradient alignment across successive mini-batches (Nichol et al., 2018; Mohtashami et al., 2022). This characteristic proves advantageous for the efficacious alternate training of multiple subnetworks within sparse co-training. The authors hence developed Alternate Sparse Training (AST) facilitating the alternate training of various subnetworks that have differing sparsity ratios. However, in our settings, which encompass both varying sparsity ratios and patterns, we found naively applying AST leads to degraded performance, owing to the strong divergence between distinct sparsity patterns.

In response, we discover that introducing a dense training phase as a "buffer" between two sparse subnetwork updates significantly attenuates this inefficacy. We term this operation the "dense pivot". Fig. 2 contrasts AST and our dense pivot co-training. Our primary intent is to harness the gradient alignment effect observed in AST and guide the gradients of disparate sparse networks. Owing to the dense pivot, the gradients of each subnetwork become "realigned" with those of the dense networks. Such alignment benefits from consistent initializations, a facet ensured by weight rewinding.

Supplementing the dense pivot, we also employ "dynamic distillation", aiming to minimize the optimization discrepancy between the dense network and its subnetworks. Assuming $\mathcal{L}(output, target)$ as the loss function and designating $i = 2, 4, 6, ...$ as the subnetwork training iterations, with $X_i$ representing the mini-batch input at the $i^{th}$ iteration and $Y_i$ signifying the ground-truth labels for $X_i$, the dynamic distillation procedure can be articulated as:

$$\mathcal{L}_{S_k} = \frac{1}{2}(\mathcal{L}(S_k(X_i), Y_i) + \mathcal{L}(S_k(X_i), \underline{\nabla}(D_k(X_i)))) \tag{1}$$

Note that $\underline{\nabla}(\cdot)$ denotes the stop gradient operator. Due to weight rewinding, all dense networks in $\{D_k\}$ are initialized identically and thus will be optimized towards similar directions although they are trained in different iterative pruning stages. Through Dense Pivot Co-training, the subnetworks $\{S_k^u, S_k^c, S_k^{nm}\}$ of different sparsity patterns are also forced to align their gradients w.r.t. $D_k$. This leads to each weight parameter being optimized in similar directions across different sparsity ratios and patterns, which contributes to the successful parameter sharing of Sparse Cocktail.

## 3.6 SPARSE NETWORK INTERPOLATION

Network merging is an emerging technique that fuses multiple neural networks into a stronger one (Nagarajan & Kolter, 2019; Frankle et al., 2020; Neyshabur et al., 2020; Von Oswald et al., 2020; Wortsman et al., 2022; Yin et al., 2022). Sparse Cocktail takes an evolving-and-merging approach: it employs an interpolation-based merging technique, fusing networks across different sparsity ratios and patterns, enhancing performance. This interpolation process is: $D_{best} = \alpha_k D_{best} + (1 - \alpha_k)D_k, k = 2, ..., N$, wherein $\alpha_k \in [0, 1]$ represents the interpolation factor, and $D_{best}$ is the preeminent interpolated network, initialized with $D_1$. The determination of $\alpha_k$ is grounded in a hold-out validation set, which we term as $ValAcc$. Following the interpolation, we refine the batch normalization statistics with a subsequent data pass, a process we denote as $BnUpdate$.

Inspired by (Wortsman et al., 2022), our method employs a meticulous, greedy strategy to discern the optimal co-efficiency within the range [0.05, 0.1, 0.2, ..., 0.9, 0.95, 1] for two subnetworks. Only subnetworks that do not diminish accuracy on the held-out set are considered for interpolation, otherwise abandoned. Post-interpolation, we implement magnitude pruning to restore the desired sparsity. The nuances of our interpolation method are detailed in Algorithm 1 in the Appendix.

Table 1: Comparison between Sparse Cocktail and other sparse co-training methods. We test ResNet-18 on CIFAR10 and ResNet-50 on ImageNet. **Co-train Patterns** mean whether the method co-trains more than one sparsity pattern (unstructured, N:M, channel-level) at once, besides the dense one. **Avg. Acc** means averaged accuracy over different sparsity ratios. To ensure a fair comparison, we implement all other methods following the original papers and test all on our pre-defined sparsity ratios. Notably, ❶ MutualNet co-trains 1 dense network and 10 channel-wise sparse networks with identical channel-wise sparsities as Sparse Cocktail at once; ❷ AC/DC co-trains 10 dense/sparse network pairs separately; ❸ AST co-trains 2 sparsity patterns separately; ❹ Sparse Cocktail co-trains all sparsity patterns and ratios **at once**.

| Method | Co-train Patterns | Sparsity Pattern | Avg. Acc(%) | | Sub-net # |
|---|---|---|---|---|---|
| | | | ResNet 18 + CIFAR10 | ResNet-50 + ImageNet | |
| MutualNet | ✗ | Dense | 92.36 | 75.94 | 1 |
| | | Channel | 90.23 | 72.04 | 10 |
| AC/DC | ✗ | Dense | **92.58** | 76.44 | 10 |
| | | Unstruct | 92.03 | **75.80** | 10 |
| AST | ✗ | Unstruct | 92.08 | 73.15 | 10 |
| | | $N{:}M$ | **92.11** | **76.02** | 4 |
| Sparse Cocktail | ✓ | Dense | 92.48 | **76.32** | 1 |
| | | Unstruct | **92.09** | 73.23 | 10 |
| | | Channel | **90.02** | 72.22 | 10 |
| | | $N{:}M$ | 91.83 | 75.19 | 3 |

# 4 EXPERIMENTS

**Dataset, Architectures, and Evaluations.** We conduct experiments on the CIFAR10 (Krizhevsky et al., 2009) and ImageNet (Deng et al., 2009) datasets. The architectures used are ResNet-18 for CIFAR10 and ResNet-50 for ImageNet (He et al., 2016). We keep the same sparsity ratios for different methods, which lead to the same inference time efficiency for each subnetwork. We evaluate the test set accuracies and parameter efficiency of the individual subnetworks, the dense networks, the total parameter number, and the subnetwork number in each shared network of each method.

**Sparse Cocktail Configurations.** For Sparse Cocktail, we record the hyperparameter setting in Table 5 in the Appendix. Additionally, we use different iterative pruning rates for the 3 sparsity patterns: for unstructured sparsity, the weight pruning rate $p_e$ is set to $0.2$ following (Frankle & Carbin, 2018); for channel-wise sparsity, the channel pruning rate $p_c$ is set to $0.1$ to keep a similar parameter number as unstructured sparsity; for $N{:}M$ sparsity, we focus on three practically accelerable sparsity ratios——1:2, 2:4 and 4:8 as in (Hubara et al., 2021). We generate these three $N{:}M$ masks during $\mathcal{P}_2, \mathcal{P}_5$ and $\mathcal{P}_8$, respectively and keep them unchanged elsewhere. Note that with UMG, the distribution of channel-wise masks will be decided by the magnitude sums of the remaining weights from both unstructured and $N{:}M$ sparsity. Under this setting, the default version of Sparse Cocktail produces **24 networks at once**, consisting of 1 dense network, 10 unstructured, 10 channel-wise and 3 $N{:}M$ subnetworks.

**Baselines and Configurations.** We compare Sparse Cocktail with three SOTA sparse co-training methods: AST (Yang et al., 2022), AC/DC (Peste et al., 2021) and MutualNet (Yang et al., 2021). Note that MutualNet uniquely includes data-level co-training by varying input image resolution; we identically followed it when using MutualNet, but did not implement the same idea in Sparse Cocktail since we want to ensure fair comparison with all other methods (without data-level co-training). The comparison of network number with shared parameters, total subnetwork number, and total parameter number of different methods is presented in Table 3. Since we enforce identical sparsity distributions for all sparse co-training methods within each sparsity pattern, the FLOPs of Sparse Cocktail within each sparsity pattern remain nearly identical with other baseline methods. We thus omit FLOPs evaluation in our experiments. More details about the implementations and hyperparameter settings are provided in Appendix B.

Table 2: The performance of individual subnetworks of different sparse co-training methods. We report 4 out of 10 evenly distributed sparsity ratios for both unstructured and channel-wise sparsities.

| Dataset | Method | | Sparsity Ratio | | | |
|---|---|---|---|---|---|---|
| | | **Unstruct** | *0.20* | *0.49* | *0.74* | *0.87* |
| | | **Channel** | *0.10* | *0.27* | *0.47* | *0.61* |
| | | $N$:$M$ | *1:1* | *1:2* | *2:4* | *4:8* |
| CIFAR10 | MutualNet | Channel | 92.32 | 91.74 | 89.54 | 87.02 |
| | AC/DC | Unstruct | 92.45 | 92.26 | 91.87 | 91.73 |
| | AST | Unstruct | 92.34 | 92.24 | 92.05 | 91.73 |
| | | $N$:$M$ | 92.56 | 92.23 | 92.45 | 92.38 |
| | Sparse Cocktail | Unstruct | 92.45 | 92.44 | 92.03 | 91.67 |
| | | Channel | 92.34 | 91.89 | 90.40 | 87.31 |
| | | $N$:$M$ | 92.48 | 92.03 | 91.93 | 91.45 |
| ImageNet | MutualNet | Channel | 75.14 | 73.62 | 71.75 | 68.32 |
| | AC/DC | Unstruct | 76.36 | 76.25 | 76.03 | 74.92 |
| | AST | Unstruct | 76.67 | 74.32 | 73.45 | 71.26 |
| | | $N$:$M$ | 76.41 | 76.07 | 75.98 | 75.61 |
| | Sparse Cocktail | Unstruct | 76.36 | 74.42 | 73.15 | 71.03 |
| | | Channel | 75.22 | 73.79 | 72.52 | 69.45 |
| | | $N$:$M$ | 76.32 | 75.23 | 74.96 | 74.23 |

Table 3: The number of networks with shared parameters, total subnetwork number (including dense) and total parameter number of each method. **Avg. Sub-net #** is obtained by dividing the network number with the sub-net number and reflects the average subnetwork capacity of each shared network, which reflects parameter efficiency for sparse co-training. Binary masks are ignored in our comparison.

| Method | Param # | Network # | Sub-net # | Avg. Sub-net # |
|---|---|---|---|---|
| MutualNet | 1x | 1 | 11 | 11 |
| AC/DC | 10x | 10 | 20 | 2 |
| AST | 1.94x | 2 | 14 | 7 |
| Sparse Cocktail | **1x** | **1** | **24** | **24** |

## 4.1 MAIN RESULTS

We collect the main experiment results in Table 1, which shows the average accuracy of different sparsity ratios for each sparsity pattern, as well as the subnetwork number for each sparsity pattern. Furthermore, we compare some individual sparsity ratio networks of different methods in Table 2.

**Comparisons with Other Co-Training Methods.** Being the most inclusive method that trains more sparse patterns or ratios than competitor methods, Sparse Cocktail achieves comparable or even better performance compared to the SOTA sparse co-training methods that only focus on one sparsity pattern per model. Specifically, ❶ when compared with AST, the current SOTA sparse-to-sparse co-training method, Sparse Cocktail achieves comparable accuracy when just comparing single sparsity pattern performance, while additionally uniting 3 sparsity patterns in one network including the channel-wise sparsity; ❷ when compared to AD/DC, Sparse Cocktail still achieves the performance on par on individual sparse patterns and ratios, while also avoiding AC/DC's need of co-training 10 dense/sparse network pairs; ❸ when compared to MutualNet which is the SOTA method on channel-wise co-training, Sparse Cocktail wins across all sparsity ratios, e.g., over $1\%$ accuracy gain at high sparsity ratios, meanwhile providing more sparsity patterns at once. Overall, Sparse Cocktail can effectively generalize and "encapsulate" these sparse co-training methods.

**Comparisons Across Architectures and Datasets.** In our main context, we conduct our experiments using ResNet-50 and image classification tasks. Now we validate that Sparse Cocktail remains effective in different architectures and tasks. Specifically, we showcase the performance of Sparse Cocktail on two network backbones——ResNet-50 and VGG-16 and on two tasks——object detection and instance segmentation on the MS COCO benchmark. We follow Table 1 for sparsity choices, *i.e.* for Sparse Cocktail we co-train 24 subnetworks with its 10 channel-wise sparsities identical to the MutualNet. We follow other experiment settings as in Section 4.4 of Yang et al. (2020) and compare the averaged performance of the 10 channel-wise subnetworks. Note that Sparse Cocktail does not use switchable resolutions as in Yang et al. (2020). The results are shown in Table 4. The results on both object detection and instance segmentation again demonstrate that Sparse Cocktail can achieve superior performance over MultualNet while co-trains 13 more unstructured and N:M subnetworks simultaneously.

## 4.2 ABLATION STUDY AND ANALYSIS

We conduct a comprehensive ablation study on Sparse Cocktail to justify the effectiveness of individual components as proposed in the methodology. Specifically, we compare our full method

Table 4: The object detection and instance segmentation results on MS COCO dataset.

| Method | Object Detection | | Instance Segmentation | |
|---|---|---|---|---|
| | BoxAP | | MaskAP | |
| | ResNet50 | VGG-16 | ResNet50 | VGG-16 |
| Baseline | 32.1 | 33.8 | 32.5 | 31.3 |
| MutualNet | **31.3** | 30.7 | 30.1 | 29.4 |
| Sparse Cocktail | 31.0 | **32.3** | **30.9** | **30.3** |

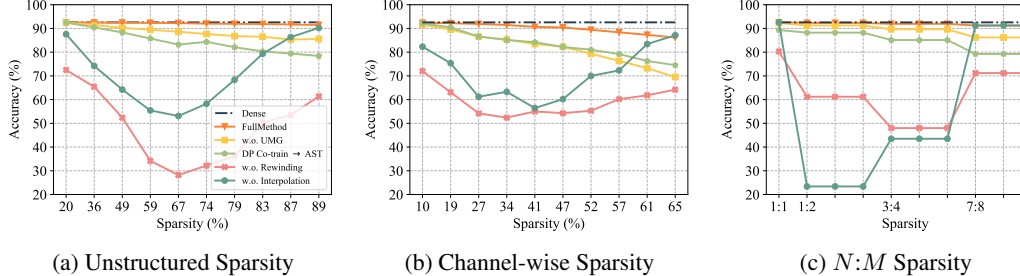

(a) Unstructured Sparsity    (b) Channel-wise Sparsity    (c) $N$:$M$ Sparsity

Figure 3: The ablation study of Sparse Cocktail. (a), (b), and (c) contain unstructured, channel-wise, and $N$:$M$ performance curves of individual sparse networks under different ablation settings. **W/o UMG** means replacing UMG with vanilla mask generation, mainly by replacing the refilling-based channel-wise mask generation with channel-weight-based mask generation. **Dense Pivot Co-training→AST** refers to replacing Dense Pivot Co-training with AST solution as proposed by (Yang et al., 2022). **W/o interpolation** means removing the network interpolation step and using the final for testing. **W/o rewinding** denotes that we immediately resume the training without rewinding the weights after pruning. Note from (c) that Sparse Cocktail only produces 3 $N$:$M$ masks.

with 4 variants ① removing UMG, ② replacing Dense Pivot Co-training with AST, ③ removing the network interpolation, and ④ removing the weight-rewinding. Moreover, we visualize the proposed interpolation process of them. The results are presented in Fig. 5.

**How does each component of Sparse Cocktail contribute?** As we can observe from Fig. 5(a), (b), (c), ❶ when we remove weight rewinding, the performance decreases drastically. This is reasonable because without weight rewinding, the dense and sparse networks at different IMP stages do not have similar optimization processes and thus end up with very different parameter values, which negatively affects network interpolation. ❷ When we remove network interpolation, we still observe a big performance drop, which highlights its importance to ensemble parameters from different IMP stages. ❸ If we replace Dense Pivot Co-training with the AST, there's also a significant performance drop, because Dense Pivot Co-training is able to regularize the subnetworks to be optimized in similar directions across different sparsity ratios, as we analyze in Section 3.5. ❹ Finally, if we remove the UMG, there is still an observable performance drop, which shows that generating masks of different sparsity patterns in a united criterion with UMG is better for sparse co-training than generating them with independent criteria. We can draw the conclusion that weight rewinding and network interpolation are both necessary components for Sparse Cocktail to function normally, while UMG and Dense Pivot Co-training also contribute remarkably to the final performance gains.

Additionally, we also investigate the role of our proposed interpolation process in Appendix H.

## 5    CONCLUSION

This paper proposes *Sparse Cocktail*, a novel method for training many resource-efficient sparse neural networks all at once. It simultaneously co-trains a diverse set of sparsity patterns, each characterized by a range of sparsity ratios. Sparse Cocktail demonstrates a competitive performance, even compared to prior single-pattern sparse co-training methods, thus generalizing and "encapsulating" those previous methods. We leave the work of exploring additional sparsity patterns to the future.

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

## A    EXISTING SPARSITY PATTERNS FOR NETWORK PRUNING

Unstructured sparsity is a technique for pruning individual weights globally but is not typically efficient on hardware. Recent developments, such as CPU-level accelerable (Kurtz et al., 2020; Liu et al., 2021c; DeepSparse, 2021) and GPU-level accelerable (Gale et al., 2020) unstructured sparsities, have been proposed to address this issue. Structured sparsity, on the other hand, is more hardware-friendly but may come at the cost of network performance. The channel-wise sparsity (Liu et al., 2017; He et al., 2017; Bartoldson et al., 2020; Rumi et al., 2020; Liu et al., 2021a) is a typical example of structured sparsity that eliminates entire channels from each layer and directly produces a slimmer network. Other examples include group-wise sparsity (Li et al., 2020; Oyedotun et al., 2020; Chen et al., 2022) that extracts sparse masks by enforcing entire rows or columns of the weight matrices to be zero, fine-grained structured sparsity (also called as $N{:}M$ sparsity) (Zhou et al., 2021; Pool & Yu, 2021; Hubara et al., 2021) that requires $N$ non-zero elements among $M$ consecutive weight parameters and can be accelerated by recent NVIDIA Ampere architecture. Other hybrid sparsity patterns, such as SIMD-friendly vector-wise sparsity (Zhou et al., 2021), pattern-based structural sparsity (Ma et al., 2020) and half-regular sparsity (Chen et al., 2018) have also been proposed to take advantage of both unstructured and structured sparsities.

## B    SPARSE NETWORK INTERPOLATION FOR SPARSE COCKTAIL

We show the network interpolation algorithm for Sparse Cocktail in Algorithm 1.

The network interpolation is used to create a single network with shared parameters among different subnetworks generated by Sparse Cocktail since we have different output parameter values of subnetworks at different iterations. It will only be performed once after the whole IMP process is finished. Our network interpolation is developed based on the interpolation method of Lottery Pool Yin et al. (2022), the core difference is that they only aim to produce a single sparse network with an interpolated sparsity ratio, while we need to use the interpolation to produce multiple subnetworks with a shared parameter set. In terms of technical details, the Lottery Pool evaluates a single network at every greedy step, while our algorithm performs interpolation using the dense networks obtained at the end of IMP iteration, and then evaluates the performance of every subnetworks obtained so far at $i$-th iteration by applying their sparse masks.

---

**Algorithm 1** Network Interpolation for Sparse Cocktail

---

1: **Input:** Dense networks $\{D_k\}$ and binary sparse masks $\{m_k^u, m_k^c, m_k^{nm}\}$ from the IMP process of Sparse Cocktail, candidate pool $C = \{0.05, 0.1, 0.2, ..., 0.9, 0.95, 1\}$ for interpolation factors, hold-out validation set $B$
2: **Output:** Interpolated dense network $D_{best}$
3: $D_{best} \leftarrow D_1$
4: **for** $k = 2$ **to** $N$ **do**
5:     $\alpha_k \leftarrow \underset{\alpha \in C}{\arg\max} \sum_{j=1}^k \sum_{m_j^u, m_j^c, m_j^{nm}} \underset{B}{\text{ValAcc}}[(\alpha D_{best} + (1-\alpha)D_k) \odot m]$
6:     $D_{best} \leftarrow \alpha_k D_{best} + (1-\alpha_k)D_k$
7: **end for**
8: BnUpdate for each subnetwork
9: **return**  $D_{best}$

---

## C    DETAILED RATIONALE ON WHY USING "UNIFIED" MASK GENERATION.

The mask generation process is called "unified" primarily because now the selection of 3 masks is all based on individual weight magnitudes by changing the pruning criterion of channel-wise pruning. In traditional channel-wise pruning, the pruning criterion is usually based on the batch norm scale factor, which is different from individual weight magnitudes. If we combine this traditional channel-wise pruning criterion with weight magnitude-based unstructured and N:M pruning, there could be conflicts for sparse co-training with different sparsity patterns. Now in our proposed unified mask generation, this is changed by introducing the refilling criterion, which decides which channels

to prune based on the magnitude sum of all individual weights in each channel. In this way, the sparse co-training can better orchestrate their shared parameters so that they will not produce conflicts in pruning criteria and cancel each other's performance.

## D  DETAILED METHODOLOGY COMPARISON OF SPARSE COCKTAIL WITH RELATED WORK

Here, we want to make some complementary discussion about some key differences and contributions of Sparse Cocktail compared to related work.

One of our major novelty is to expand the scope of sparse co-training to cover diverse sparsity patterns and multiple sparsity ratios at once. This research goal stands out as novel because previous works have not addressed such a wide range of sparse patterns and ratios. The harmony among different sparsity patterns is made possible by UMG and Dense Pivot Co-training. By using UMG as a universal pruning criterion and producing closely aligned sparse masks, we relieve the gradient conflicts of different sparsity patterns during training. Then by Dense Pivot Co-training that inserts a dense mini-batch step at every alternative sparse mini-batch step, we further enforce the optimizing directions of subnetworks of different sparsity patterns to be aligned with the same dense network. Meanwhile, the dense network at each IMP iteration has the same initialization as in the Lottery Ticket Hypothesis, thus the optimization directions from different IMP iteration are aligned because of the same dense network initialization. Thus, the optimization directions from different sparsity ratios and patterns are all regularized to be aligned together. We are also the first one to apply LTH for sparse co-training to amortize the sparse co-training pressure (we only co-trains subnetworks of a single sparsity ratio from different sparsity patterns at the same time in each IMP iteration, while finally it produces a lot more subnetworks in total without the need to co-train them together), while related work such as Lottery Pool only considers aiming to produce a single stronger subnetwork.

We also do not simply reuse existing methods but develop novel adaptations for the sparse co-training circumstance. Specifically, (1) we adapt the refilling method in Chen et al. (2022) as UMG by incorporating N:M sparsity and letting both unstructured and N:M sparsity decide which channels to refill; (2) we adapt the Yang et al. (2022) as Dense Pivot Co-training by not just alternating mini-batches among sparse networks of the same sparsity ratio and pattern but inserting a dense mini-batch step and combine it with IMP to achieve optimization alignments across different sparsity ratios and patterns. (3) we adapt the network interpolation method in Yin et al. (2022) as we state in A6 below.

## E  HYPERPARAM SETTING OF SPARSE COCKTAIL

We show our hyperparameter settings of Sparse Cocktail in Table 5 for CIFAR10 dataset and Table 6 for ImageNet dataset, respectively.

## F  IMPLEMENTATIONS AND SETTINGS OF THE BASELINE METHODS

For AST (Yang et al., 2022),we implement the gradient correction and inner-group iteration as in (Yang et al., 2022). Following (Yang et al., 2022), and co-train 10 unstructured or 4 $N$:$M$ subnetworks in one experiment, with the same number of total training epochs as Sparse Cocktail (1500). For AC/DC (Peste et al., 2021), we co-train one unstructured subnetwork with the dense network in one experiment with 150 training epochs per experiment. For MutualNet (Yang et al., 2021), we also use 1500 training epochs to co-train 10 channel-wise subnetworks, and uses switachble input resolutions of 32, 28, 24, 16 for CIFAR10 and 224, 196, 160, 128 for ImageNet, repsectively. All baseline methods are trained to produce the same sparsity ratios as the Sparse Cocktail within a single sparsity pattern. We keep all the other hyperparameters the same as Sparse Cocktail among all baseline methods. Early stop is used to avoid overfitting for all methods.

Based on the above settings, all the methods have the same number of training iterations and batch size (regardless of which subnetwork will be trained at each mini-batch), and thus the same training

---

[1]https://github.com/abhuse/cyclic-cosine-decay

Table 5: The hyperparameter setting of Sparse Cocktail on CIFAR10 dataset.

| Hyperparameter | Configuration |
|---|---|
| IMP setting | iterations = 10, rewind epoch = 7, training epochs = 150 |
| Optimizer | SGD (lr = 0.1, momentum = 0.9, weight_decay = $1e-4$) |
| LR scheduler | CyclicCosineDecay[1](init_epoch = 100, interval = 10, min_lr = 0.001, restart_lr = 0.01) |
| Candidate Pool $B$ | {0.05, 0.1, 0.2, ..., 0.9, 0.95, 1} |
| Hold-out validation set | Last 5% of training set |
| Pruning Range | All convolutional layers except the 1st layer |
| Pruning Rate | 0.2 for unstructured 0.1 for channel-wise 1:2, 2:4, 4:8 for $N$:$M$ |

Table 6: The hyperparameter setting of Sparse Cocktail on ImageNet dataset.

| Hyperparameter | Configuration |
|---|---|
| IMP setting | iterations = 10, rewind epoch = 5, training epochs = 90 |
| Optimizer | SGD (lr = 0.1, momentum = 0.9, weight_decay = $1e-4$) |
| LR scheduler | CyclicCosineDecay(init_epoch = 70, interval = 10, min_lr = 0.001, restart_lr = 0.01) |
| Candidate Pool $B$ | {0.05, 0.1, 0.2, ..., 0.9, 0.95, 1} |
| Hold-out validation set | Last 5% of training set |
| Pruning Range | All convolutional layers except the 1st layer |
| Pruning Rate | 0.2 for unstructured 0.1 for channel-wise 1:2, 2:4, 4:8 for $N$:$M$ |

cost. We also empirically find that Sparse Cocktail has only around 1/8 extra total wall-clock training time primarily due to 1 extra distillation step every 2 mini-batches.

## G  THE INFLUENCE OF HIGHER SPARSITY ON SPARSE CO-TRAINING WITH DIFFERENT SPARSITY RATIOS

Under our hyper-parameter setting, the unstructured and structured sparsity ratios range in $[0.20, 0.87]$ and $[0.10, 0.61]$, respectively. We do not use higher sparsity ratios higher than 90% primarily because the compared sparse co-training methods all have performance degradation when the sparsity gets very high. In the two compared unstructured sparse co-training methods, AC/DC Peste et al. (2021) has 3.5% performance degradation (compared to vanilla dense network) at 95% sparsity and 8.5% degradation at 98% sparsity; AST Yang et al. (2022) has less performance degradation at high sparsity primarily likely because it doesn't involve co-training with different sparsity ratios and only focus on single sparsity ratio but different masks.

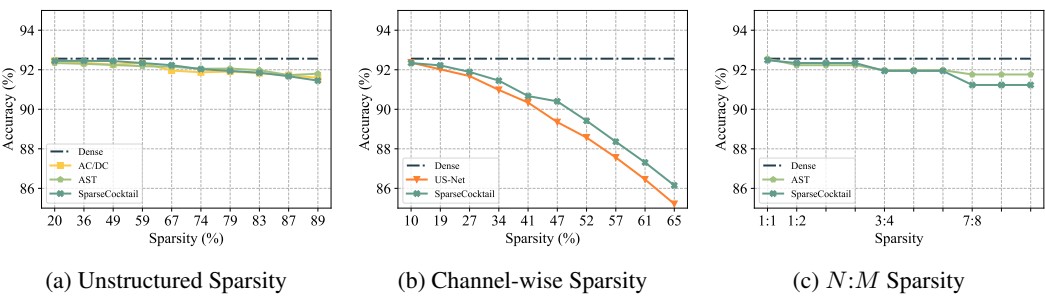

Figure 4: The performance comparison of individual sparse networks of different sparse co-training methods on CIFAR10 dataset.

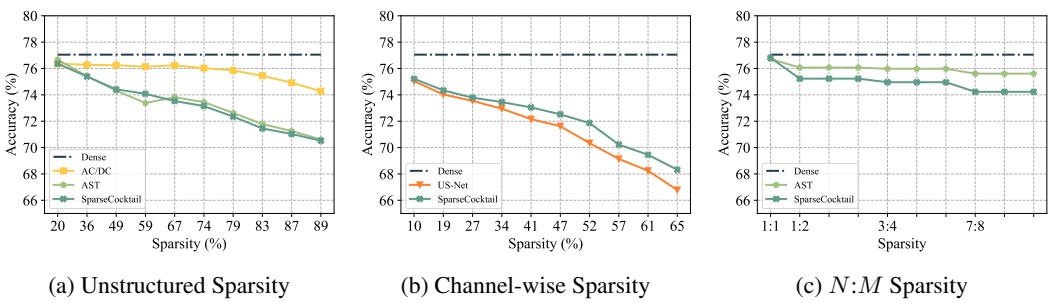

Figure 5: The performance comparison of individual sparse networks of different sparse co-training methods on the ImageNet dataset.

## H   WHAT IS THE ROLE OF THE PROPOSED INTERPOLATION PROCESS?

Recall that Algorithm 1 greedily chooses interpolation factors $\alpha_k$ as k increases based on the average accuracy of all the subnetworks currently being considered, i.e. $\{S_j | j = 1, 2, ..., k\}$. From Fig. 6, we find that our full method has a reasonable interpolation process. Compared to the **Dense Pivot Co-training→AST** setting that simply averages all the networks from different IMP stages, and the **W/o rewinding** setting where the interpolation is severely biased towards the parameters at the last several IMP stages, our full method reaches a good balance between these two settings.

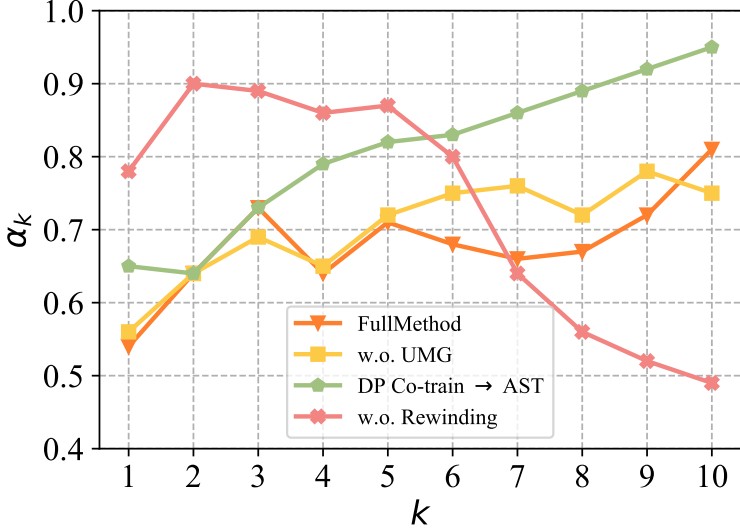

Figure 6: The changes of the optimal interpolation factors $\alpha_k$ as $k$ increases in Algorithm 1.

Table 7: The performance influence of network interpolation in Sparse Cocktail.

| Dataset | CIFAR10 | | ImageNet | |
|---|---|---|---|---|
| | Before Interp. | After Interp. | Before Interp. | After Interp. |
| Dense | 92.56 | 92.48 | 76.45 | 76.32 |
| Unstructured | 92.45 | 92.09 | 74.64 | 73.23 |
| Channel-wise | 90.97 | 90.02 | 74.02 | 72.22 |
| N:M | 92.32 | 91.83 | 76.13 | 75.19 |

## I   THE PERFORMANCE INFLUENCE OF NETWORK INTERPOLATION

Regarding the performance influence of network interpolation, we list the average performance before and after the interpolation of Sparse Cocktail on image classification datasets in Table 7. The results show that there will be slight performance degradation after interpolation, but we consider this a necessary sacrifice since we aim to produce a single network with shared parameters of multiple subnetworks.

We empirically find that different subnetworks of the same sparsity pattern obtained by IMP with weight rewinding are located in the same loss basin, i.e. there are no significant error barriers for interpolations similar to Figure 2 in Yin et al. (2022), while subnetworks from different sparsity patterns have at most 3.2% error barriers on average due to the divergence in sparse masks. We show the latter phenomenon in Fig. 7, by plotting the performance change using different interpolation factors as in Algorithm 1. However, by finding proper interpolations the error barrier problem can be mitigated or avoided.

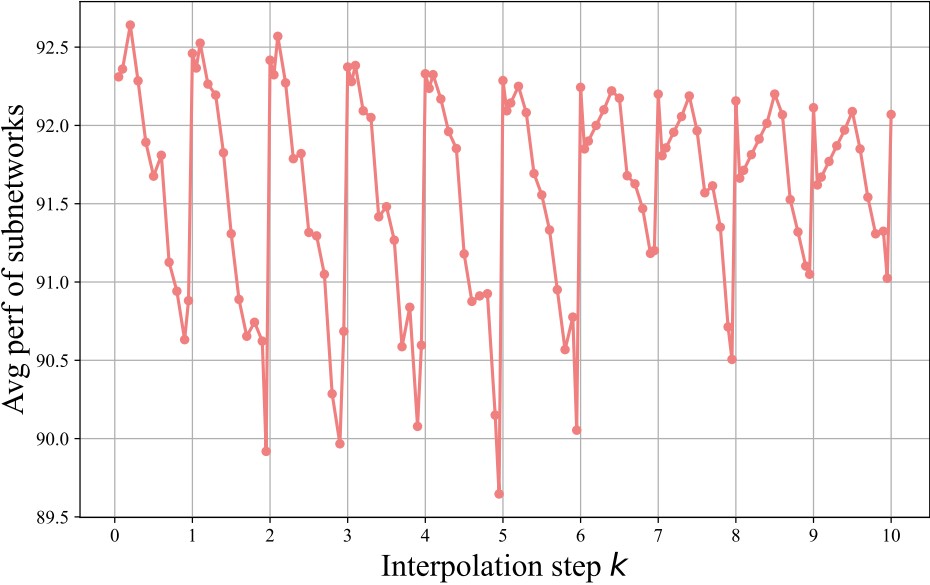

Figure 7: The averaged candidate interpolation performance of all subnetworks considered at greedy step $k$ as in Algorithm 1. At each greedy interpolation step $k$, we use $D_{best}$ (that represents the shared dense network of the best average performance of subnetworks obtained before $k$-th IMP iteration) and $D_k$ to perform the interpolation, and $[k, k+1)$ represent the interpolated average performance of all subnetworks obtained before $k+1$-th IMP iteration, where the points of $x$ value between $[k, k+1)$ represent the interpolation factor $1 - \alpha_k$.

