# OpenReview forum: "Sparse Cocktail: Every Sparse Pattern Every Sparse Ratio All At Once"
_ICLR.cc/2024/Conference — Submitted to ICLR 2024_

### Official Review · Reviewer_1N1a · 2023-10-30

**Soundness:** 3 good
**Presentation:** 3 good
**Contribution:** 3 good
**Rating:** 6
**Confidence:** 4

**Summary:**

Sparse Cocktail is a novel sparse co-training framework that can concurrently produce multiple sparse subnetworks across a spectrum of sparsity patterns and ratios, in addition to a dense model.

**Strengths:**

Key technical contributions include:

(S1) Simultaneously co-trains diverse sparsity patterns (unstructured, channel-wise, N:M) each with multiple sparsity ratios. The well-articulate problem is an important strength.
(S2) Uses iterative pruning with weight rewinding to segregate subnetworks of different sparsity ratios
(S3) Proposes a Unified Mask Generation technique to jointly produce masks of different patterns
(S4) Employs Dense Pivot Co-training to align optimization of diverse sparse subnetworks
(S5) Performs Sparse Network Interpolation to further boost performance (relatively old trick)

Key experimental strengths include:

(S6) Sparse Cocktail achieves comparable or better performance than SOTA sparse co-training methods that focus on single patterns only. It generalizes previous methods while producing more subnetworks at once. Its performance can be on par with or even better than strong baselines such as AST and MutualNet.
(S7) Besides evaluation on CIFAR10/ImageNet with ResNet/VGG, it also transfers effectively to object detection and instance segmentation tasks.
(S8) In ablation studies, key components like weight rewinding, network interpolation, Unified Mask Generation and Dense Pivot Co-training are shown to contribute to Sparse Cocktail's performance

**Weaknesses:**

(W1) The whole pipeline looks like a huge ensemble of existing techniques, such as the "Dense Pivot Co-training" stage from USNet and BigNAS, the "Sparse Network Interpolation" stage from AutoSlim and LotteryPool … However, the author did not make meaningful discussions in each stage, on their differences from prior arts. I would like to hear the authors clarify.

(W2) I would like to see some more relevant metrics such as training time, memory savings, or inference speed ups if any. Without those, it is hard or meaningless to fetch any real benefit of training with sparsity.

(W3) Is Dense Pivot Co-training just weight rewinding (which is a pretty standard trick), or are they different (in which way)?

(W4) Why the three mask generations in Section 3.4. are called “unified”?

**Questions:**

See W1-W4

---

> ### Author Response · Authors · 2023-11-22
> **Rebuttal for Reviewer 1N1a's review**
>
> Dear reviewer,
>
> Thank you for the time and effort you have invested in reviewing our manuscript. We appreciate your valuable comments and suggestions. Below, we address each of your points in detail.
>
> >Q1:  The whole pipeline looks like a huge ensemble of existing techniques, such as the "Dense Pivot Co-training" stage from USNet and BigNAS, the "Sparse Network Interpolation" stage from AutoSlim and LotteryPool … However, the author did not make meaningful discussions in each stage, on their differences from prior arts. I would like to hear the authors clarify.
>
> A1: We would like to argue that one of our major novelty is to expand the scope of sparse co-training to cover diverse sparsity patterns and multiple sparsity ratios at once. This research goal stands out as novel because previous works have not addressed such a wide range of sparse patterns and ratios. The harmony among different sparsity patterns is made possible by UMG and Dense Pivot Co-training. By using UMG as a universal pruning criterion and producing closely aligned sparse masks, we relieve the gradient conflicts of different sparsity patterns during training. Then by Dense Pivot Co-training that inserts a dense mini-batch step at every alternative sparse mini-batch step, we further enforce the optimizing directions of subnetworks of different sparsity patterns to be aligned with the same dense network. Meanwhile, the dense network at each IMP iteration has the same initialization as in the Lottery Ticket Hypothesis, thus the optimization directions from different IMP iteration are aligned because of the same dense network initialization. Thus, the optimization directions from different sparsity ratios and patterns are all regularized to be aligned together. We are also the first one to apply LTH for sparse co-training to amortize the sparse co-training pressure (we only co-trains subnetworks of a single sparsity ratio from different sparsity patterns at the same time in each IMP iteration, while finally it produces a lot more subnetworks in total without the need to co-train them together.), while related work such as Lottery Pool only considers aiming to produce a single stronger subnetwork.
>
>
> We also do not simply reuse existing methods but develop novel adaptations for the sparse co-training circumstance. Specifically, (1) we adapt the refilling method in Chen et al[1] as UMG by incorporating N:M sparsity and letting both unstructured and N:M sparsity decide which channels to refill; (2) we adapt the AST[2](instead of USNet or BigNAS) as Dense Pivot Co-training by not just alternating mini-batches among sparse networks of the same sparsity ratio and pattern but inserting a dense mini-batch step and combine it with IMP to achieve optimization alignments across different sparsity ratios and patterns. (3) we adapt the network interpolation method in Lottery Pool[2] as our network interpolation for sparse co-training and the core difference is that they only aim to produce a single sparse network with an interpolated sparsity ratio, while we need to use the interpolation to produce multiple subnetworks with a shared parameter set.
>
> >Q2: I would like to see some more relevant metrics such as training time, memory savings, or inference speed ups if any. Without those, it is hard or meaningless to fetch any real benefit of training with sparsity.
>
> A2: Thanks for the question. we want to emphasize that the motivation of our sparse co-training is parameter efficiency instead of training or inference speed up. Since for each kind of sparsity pattern, the Sparse Cocktail produces multiple subnetworks with exactly the same sparsity ratios w.r.t. other corresponding methods, we thus have almost the same memory savings, and inference speed up and omit the comparison. We also empirically find that Sparse Cocktail has only around 1/8 extra total wall-clock training time primarily due to 1 extra distillation step every 2 mini-batches.

---

> ### Author Response · Authors · 2023-11-22
> **Follow-up response**
>
> >Q3: Is Dense Pivot Co-training just weight rewinding (which is a pretty standard trick), or are they different (in which way)?
>
> A3: The dense pivot co-training is an orthogonal technique w.r.t. the weight-rewinding. The weight-rewinding means rewinding the weights to initialization weights(or very early epochs) at the start of each IMP iteration, which is operated on the epoch level. For example, if the IMP has 10 iterations and each iteration trains the network for 100 epochs, then the weight rewinding will be performed for 9 times at the start of each iteration, from the 2nd to the last iteration.
> In contrast, the Dense Pivot Co-training inserts a dense network training step before each mini-batch of the sparse co-training, where parameters are shared and continuously updated without rewinding and it is operated on a mini-batch level. For example, at the i-th iteration of Sparse Cocktail, the mini-batch orders during training are arranged like this: dense —> unstructured subnetwork —> dense  —> channel-wise subnetwork  —> dense  —> N:M subnetwork  —> dense  —> unstructured subnetwork ...
> To summarize, weight rewinding is an epoch-level operation and dense-pivot co-training is a mini-batch-level operation, and they are thus orthogonal.
>
>
> >Q4: Why the three mask generations in Section 3.4. are called “unified”?
>
>
> A4: The mask generation is called "unified" primarily because now the selection of 3 masks are all based on individual weight magnitudes by changing the pruning criterion of channel-wise pruning. In traditional channel-wise pruning, the pruning criterion is usually based on the batch norm scale factor, which is different from individual weight magnitudes. If we combine this traditional channel-wise pruning criterion with weight magnitude-based unstructured and N:M pruning, there could be conflicts for sparse co-training with different sparsity patterns. Now in our proposed unified mask generation, this is changed by introducing the refilling criterion, which decides which channels to prune based on the magnitude sum of all individual weights in each channel. In this way, the sparse co-training can better orchestrate their shared parameters so that they will not produce conflicts in pruning criteria and cancel each other's performance.

---

### Official Review · Reviewer_TUJX · 2023-10-30

**Soundness:** 3 good
**Presentation:** 3 good
**Contribution:** 3 good
**Rating:** 8
**Confidence:** 4

**Summary:**

This paper proposed a new joint sparse training algorithm called “Sparse Cocktail”, that allows for the selection of the desired sparsity pattern and ratio at inference. The benefits of using Sparse Cocktail for training sparse neural networks include the ability to produce a diverse set of sparse subnetworks with various sparsity patterns and ratios at once, making it easier to switch between them depending on hardware availability.

**Strengths:**

Overall, Sparse Cocktail can effectively generalize and encapsulate previous sparse co-training methods. Experiment results look promising, and paper writing is clear to follow (plus a lovely title :)
In more details:

-	Sparse Cocktail differs from other sparse co-training approaches in that it can produce multiple sparse subnetworks across a spectrum of sparsity patterns and ratios simultaneously, while previous approaches only focus on one or two types of sparsity patterns and/or with different sparsity ratios.
-	The approach alternates between various sparsity pattern training phases, incrementally raising the sparsity ratio across these phases. Underlying the multi-phase training is a unified mask generation process that allows seamless phase transitions without performance breakdown.
-	The authors also complement a dense pivot co-training strategy augmented with dynamic distillation, aligning the optimization trajectories of diverse sparse subnetworks. In the end, all sparse subnetworks share weights from the dense network, culminating in a "cocktail" of dense and sparse models, offering a highly storage-efficient ensemble.
-	The paper shows that Sparse Cocktail achieves great parameter efficiency and comparable Pareto-optimal trade-off individually achieved by other sparse co-training methods. Sparse Cocktail achieves comparable or even better performance compared to the state-of-the-art sparse co-training methods that only focus on one sparsity pattern per model. Additionally, Sparse Cocktail avoids the need for co-training multiple dense/sparse network pairs, making it a more storage-efficient ensemble.

**Weaknesses:**

•	No discussion of training time cost. The proposed joint/switchable training appears to take much longer time than any single sparse training method. Please report the details and provide a fair discussion on training cost.
•	Hyperparameter setting was missed in Appendix C (empty - though mentioned multiple times in the main paper)!! This paper has so many moving widgets and it seems challenging to get all the hyper-parameters and settings right in practice.

**Questions:**

Overall the paper is clear, but several important pieces of information were missed, as pointed out in the weakness part.

---

> ### Author Response · Authors · 2023-11-22
> **Rebuttal for Reviewer TUJX's review**
>
> Dear reviewer,
>
> Thank you for the time and effort you have invested in reviewing our manuscript. We appreciate your valuable comments and suggestions. Below, we address each of your points in detail.
>
>
> >Q1: No discussion of training time cost. The proposed joint/switchable training appears to take much longer time than any single sparse training method. Please report the details and provide a fair discussion on training cost.
>
> A1: Thanks for your question. We have discussed the training cost in Appendix D and will make this point more clear in the revised version. Overall, we keep the batch size and total training epochs of all the methods, including Sparse Cocktail, AST, AC/DC, and MutualNet, as 1500 epochs to have a fair comparison. Specifically, (1) for Sparse Cocktail, we use 10 iterations for IMP and each iteration contains 150 epochs. (2) for AST and MutualNet, we directly perform the co-training for 1500 epochs. (3) for AC/DC we co-train each of the 10 dense-sparse network pairs for 150 epochs. In this way, all the methods have the same number of training iterations and batch size (regardless of which subnetwork will be trained at each mini-batch), and thus the same training cost.  We also empirically find that Sparse Cocktail has only around 1/8 extra total wall-clock training time primarily due to 1 extra distillation step every 2 mini-batches.
>
>
> >Q2: Hyperparameter setting was missed in Appendix C (empty - though mentioned multiple times in the main paper)!! This paper has so many moving widgets and it seems challenging to get all the hyper-parameters and settings right in practice.
>
> A2: We apologize for this problem, we have presented the hyper-parameters in Table 5 and Table 6 in the Appendix, but this is misplaced away from the Appendix C section title due to the typesetting problem. We will fix this issue and add textual references in Appendix C in the revised version.

---

### Official Review · Reviewer_MPL5 · 2023-10-31

**Soundness:** 2 fair
**Presentation:** 2 fair
**Contribution:** 2 fair
**Rating:** 3
**Confidence:** 3

**Summary:**

This paper aims at performing sparse cotraining to obtain multiple sparse networks at once with different sparsity ratios and sparsity types (unstructured, structured or N:M). The authors propose to use a combination of iterative magnitude pruning, unifying masks and interspersed dense training in order to obtain multiple subnetworks within the same network for different sparsity ratios and sparsity types.

**Strengths:**

The authors present a sparse cotraining method that can obtain subnetworks of different sparsity ratios and sparsity types at once.

**Weaknesses:**

I am concerned about the novel contributions of this paper, and the results presented in this paper are the combination of existing works with little novelty of its own.

1. The results are shown on different sparse subnetworks obtained from multiple sparse masks. However, it is likely that the performance of these sparse subnetworks is stable merely because of the relatively low sparsity reported in the paper. In order to see the effectiveness of the method, I would like to see the performance of the subnetworks with higher sparsity (> 90%) especially for unstuctured sparsity patterns.

2. The algorithm is not entirely clear from the Figure and methodology section. For example, how many sparsities is each sparse pattern trained for, what are the performances of each sparsity pattern and how does a subnetwork’s performance improve after merging (if it does).

3. The author’s don’t comment on the loss landscape of each of the subnetworks obtained during training. From previous work by Paul et al [1] I would expect each of the obtained subnetworks to lie in the same loss basin. In order to assess the effectiveness of the dynamic distillation step I would expect to look at the Hessian or the linear mode connectivity between the subnetworks obtained.

4. Additionally, the performance of the proposed method on ImageNet is poorer than AC/DC (in Table 1) which is a well established method.

Overall my primary concern is that the novelty of this paper is limited as the authors have put together multiple existing methods (AST, AC/DC) in order to obtain multiple subnetworks at once.
However, the attained subnetworks themselves have not been confirmed to be effective at higher sparsities.

[1] Paul, Mansheej, et al. "Unmasking the Lottery Ticket Hypothesis: What's Encoded in a Winning Ticket's Mask?." International Conference on Learning Representations 2022.

**Questions:**

1. How does Network Interpolation help, and at what stage of training is it used. It seems to be similar to the implementation of Lottery Pools [1].

2. Its not made clear how the N:M network and Unstructured networks obtained from IMP are kept similar to each other such that their weights can be interpolated.

3. It is not clear to me why the authors choose to generate a total of 24 subnetworks by restricting the unstrcutured and structured sparse networks to 10 each. Is this a hyperparameter and why not choose additional networks at higher sparsity ratios?

[1] Yin, Lu, et al. "Lottery pools: Winning more by interpolating tickets without increasing training or inference cost." Proceedings of the AAAI Conference on Artificial Intelligence 2023.

---

> ### Author Response · Authors · 2023-11-22
> **Rebuttal to Reviewer MPL5's review**
>
> Dear reviewer,
>
> Thank you for the time and effort you have invested in reviewing our manuscript. We appreciate your valuable comments and suggestions. Below, we address each of your points in detail.
>
> >Q1: I am concerned about the novel contributions of this paper, and the results presented in this paper are a combination of existing works with little novelty of its own.
>
> A1: We would like to argue that one of our major novelty is to expand the scope of sparse co-training to cover diverse sparsity patterns and multiple sparsity ratios at once. This research goal stands out as novel because previous works have not addressed such a wide range of sparse patterns and ratios. The harmony among different sparsity patterns is made possible by UMG and Dense Pivot Co-training. By using UMG as a universal pruning criterion and producing closely aligned sparse masks, we relieve the gradient conflicts of different sparsity patterns during training. Then by Dense Pivot Co-training that inserts a dense mini-batch step at every alternative sparse mini-batch step, we further enforce the optimizing directions of subnetworks of different sparsity patterns to be aligned with the same dense network. Meanwhile, the dense network at each IMP iteration has the same initialization as in the Lottery Ticket Hypothesis, thus the optimization directions from different IMP iteration are aligned because of the same dense network initialization. Thus, the optimization directions from different sparsity ratios and patterns are all regularized to be aligned together. We are also the first one to apply LTH for sparse co-training to amortize the sparse co-training pressure (we only co-trains subnetworks of a single sparsity ratio from different sparsity patterns at the same time in each IMP iteration, while finally it produces a lot more subnetworks in total without the need to co-train them together), while related work such as Lottery Pool only considers aiming to produce a single stronger subnetwork.
>
> We also do not simply reuse existing methods but develop novel adaptations for the sparse co-training circumstance. Specifically, (1) we adapt the refilling method in Chen et al[1] as UMG by incorporating N:M sparsity and letting both unstructured and N:M sparsity decide which channels to refill; (2) we adapt the AST[2] as Dense Pivot Co-training by not just alternating mini-batches among sparse networks of the same sparsity ratio and pattern but inserting a dense mini-batch step and combine it with IMP to achieve optimization alignments across different sparsity ratios and patterns. (3) we adapt the network interpolation method in Lottery Pool[3] as we state in A6 below.
>
>
> >Q2: The results are shown on different sparse subnetworks obtained from multiple sparse masks. However, it is likely that the performance of these sparse subnetworks is stable merely because of the relatively low sparsity reported in the paper. In order to see the effectiveness of the method, I would like to see the performance of the subnetworks with higher sparsity (> 90%) especially for unstructured sparsity patterns.
>
> A2: We would like to argue the compared sparse co-training methods all have performance degradation when the sparsity gets very high. In the two compared unstructured sparse co-training methods, AC/DC [4] has ~3.5% performance degradation (compared to vanilla dense network) at 95% sparsity and ~8.5% degradation at 98% sparsity; AST [2] has less performance degradation at high sparsity primarily likely because it doesn't involve co-training with different sparsity ratios and only focus on single sparsity ratio but different masks.

---

> > ### Author Response · Authors · 2023-11-22
> > **Follow-up rebuttal**
> >
> > >Q5: Additionally, the performance of the proposed method on ImageNet is poorer than AC/DC (in Table 1) which is a well established method.
> >
> > A5: We would like to argue that AC/DC has slightly better performance mainly because the original AC/DC algorithm only co-trains 2 networks (one dense and one sparse) at the same time. In our experiments, we compare Sparse Cocktail, which co-trains 24 subnetworks with different sparsity patterns and ratios at once, with AC/DC which co-trains 2 networks for 10 separate times to produce 10 unstructured subnetworks without parameter sharing. Our Sparse Cocktail achieves much higher parameter efficiency than AC/DC (24 vs 2 avg subnetwork number as in Table 3) with slight sacrifices in the performance of individual subnetworks (which we believe is more a merit than a downside).
> >
> > >Q6: How does Network Interpolation help, and at what stage of training is it used. It seems to be similar to the implementation of Lottery Pools [1].
> >
> > A6: The network interpolation is used to create a single network with shared parameters among different subnetworks generated by Sparse Cocktail since we have different output parameter values of subnetworks at different iterations. It will only be performed once after the whole IMP process is finished. Our network interpolation is developed based on the interpolation method of Lottery Pool, the core difference is that they only aim to produce a single sparse network with an interpolated sparsity ratio, while we need to use the interpolation to produce multiple subnetworks with a shared parameter set. In terms of technical details, the Lottery Pool evaluates a single network at every greedy step, while our algorithm performs interpolation using the dense networks obtained at the end of IMP iteration (please refer to Appendix B for algorithm details), and then evaluates the performance of every subnetworks obtained so far at i-th iteration by applying their sparse masks.
> >
> > >Q7: It's not made clear how the N:M network and Unstructured networks obtained from IMP are kept similar to each other such that their weights can be interpolated.
> >
> > A7: Thanks for the question. The weight interpolation among different sparsity patterns is made possible by UMG and Dense Pivot Co-training. By using UMG as a universal pruning criterion, we relieve the gradient conflicts of different sparsity patterns during training. Then by Dense Pivot Co-training, we further enforce the optimizing directions of subnetworks of different sparsity patterns to be aligned with the dense network. Meanwhile, the dense network at each IMP iteration has the same initialization as in Lottery Ticket Hypothesis, thus the optimization directions from different IMP iterations are aligned because of the same dense network initialization. Thus, overall weight interpolation from different sparsity ratios and patterns is possible.
> >
> > >Q8: It is not clear to me why the authors choose to generate a total of 24 subnetworks by restricting the unstructured and structured sparse networks to 10 each. Is this a hyper-parameter and why not choose additional networks at higher sparsity ratios?
> >
> > A8: Please refer to the response in A3. This is an adjustable hyperparameter and we elaborate on how we choose to generate a total of 24 subnetworks at the start of Section 4. As we discussed in A3, co-training at higher sparsity ratios along with a dense network will cause performance degradation problems not only for Sparse Cocktail but also for the compared methods.
> >
> >
> >
> > [1] Coarsening the Granularity: Towards Structurally Sparse Lottery Tickets. Tianlong Chen et al. 2022.
> >
> > [2] Get More at Once: Alternating Sparse Training with Gradient Correction. Li Yang et al. 2022.
> >
> > [3] Lottery Pools: Winning More by Interpolating Tickets without Increasing Training or Inference Cost. Yin Lu et al. 2022.
> >
> > [4] AC/DC: Alternating Compressed/DeCompressed Training of Deep Neural Networks. Alexandra Peste et al. 2021.

---

> > > ### Comment · Reviewer_MPL5 · 2023-11-23
> > > **Response to Rebuttal**
> > >
> > > I thank the authors for providing comprehensive clarifications. Their response significantly makes clear a lot of details about the training method of Sparse Cocktail.
> > >
> > > However, I am still unconvinced of the novel contributions of the paper. It seems that the main goal still is to combine multiple existing methods by extensive hyperparameter tuning and engineering like Dense Co-pivot training.
> > >
> > > Moreover, only by training IMP, I could potentially obtain multiple sparse networks by removing $20$% of the nonzero parameters in each prune-train iteration of density $0.8, 0.8^2, 0.8^3, ...$ and so on.
> > > In order to achieve a final sparsity of $0.8^l$, one would need to train all the previous sparsity levels ${1, 2, .., l-1}$, thus generating multiple sparse networks with increasing sparsity ratios. Each of these networks could then be transferred to a structured sparse network by using the work of Chen et al. [1]. And a similar approach could be followed for N:M sparsity.
> > > I don't see the benefit of combining these multiple methods at the cost of extensive hyperparameter tuning and significantly increased training time to achieve multiple sparse patterns.
> > >
> > > Hence, I would still lean on rejection and keep my score.
> > >
> > > [1] Chen, Tianlong, et al. "Coarsening the granularity: Towards structurally sparse lottery tickets." International Conference on Machine Learning. PMLR, 2022.

---

> ### Author Response · Authors · 2023-11-22
> **follow-up rebuttal**
>
> >Q3: The algorithm is not entirely clear from the Figure and methodology section. For example, how many sparsities is each sparse pattern trained for, what are the performances of each sparsity pattern and how does a subnetwork’s performance improve after merging (if it does).
>
> A3: Thanks for your question. The number of sparsity ratios can be a variable hyper-parameter while in our setting, we use 10 sparsity ratios for both unstructured and channel-wise sparsity, and 3 sparsity ratios for N:M sparsity (1:2, 2:4, 4:8) following prior sparsity work as we discussed in the start of Section 4. The performance of each sparsity pattern are shown in Table 1 and Table 2, generally speaking, unstructured and N:M subnetworks have comparable performance to the dense network and channel-wise subnetworks have degraded performance due to the structural sparsity nature. Regarding the performance influence of network interpolation, we list the average performance before and after Interpolation of Sparse Cocktail on image classification datasets. The results show that there will be slight performance degradation after interpolation, but we consider this a necessary sacrifice since we aim to produce a single network with shared parameters of multiple subnetworks.
>
>
> On CIFAR10 dataset:
> |  Sparsity Pattern  |   Dense |   Unstructured|  Channel-wise | N:M |
> |-------------:|----------:|-----------:|-----------:|-----------:|
> |Before Interpolation |  92.56 |   92.45 |   90.97  |    92.32  |
> |After Interpolation |    92.48 |   92.09 |   90.02  |    91.83  |
>
>
> On ImageNet dataset:
> |   Sparsity Pattern |   Dense |   Unstructured|  Channel-wise | N:M |
> |-------------:|----------:|-----------:|-----------:|-----------:|
> |Before Interpolation | 76.45  |   74.64 |   74.02 |    76.13  |
> |After Interpolation |   76.32  |   73.23 |   72.22  |    75.19  |
>
>
> >Q4: The author’s don’t comment on the loss landscape of each of the subnetworks obtained during training. From previous work by Paul et al [1] I would expect each of the obtained subnetworks to lie in the same loss basin. In order to assess the effectiveness of the dynamic distillation step I would expect to look at the Hessian or the linear mode connectivity between the subnetworks obtained.
>
> A4: Thanks for your comment. We empirically find that different subnetworks of the same sparsity pattern obtained by IMP with weight rewinding are located in the same loss basin, i.e. there are no significant error barriers for interpolations similar to Figure 2 in [3], while subnetworks from different sparsity patterns have at most ~3.2% error barriers on average due to the divergence in sparse masks. We show the latter phenomenon in Figure 7 in appendix, by plotting the performance change using different interpolation factors as in Algorithm 1. However, by finding proper interpolations the error barrier problem can be mitigated or avoided.

---

> ### Author Response · Authors · 2023-11-23
> **Reponse to reviewer MPL5's further comments**
>
> Dear Reviewer,
>
> Thanks for your further questions. We can address your concerns as follows:
>
> >Q9: However, I am still unconvinced of the novel contributions of the paper. It seems that the main goal still is to combine multiple existing methods by extensive hyperparameter tuning and engineering like Dense Co-pivot training.
>
> A9: As we have discussed in Section 1, our main goal is to train multiple sparsity ratios and patterns all at once, in order to achieve real-time switching with the need to store only one set of shared parameters, based on the inference platforms that may have different supports on accelerable sparsity patterns and their available resources may vary over time which needs switchable sparsity ratios.
>
> We do not include extensive hyperparameter tuning. As we show in Table 5 and 6 in the Appendix, we just follow the default hyperparameters as in the original LTH paper [5] for producing unstructured lottery tickets, and keep the element-wise sparsity roughly the same  for channel-wise sparsity pattern by setting channel-wise pruning ratio as 0.1. The candidate pool is also not a hyperparameter but a search space of our greedy network interpolation algorithm. Other optimization hyperparameters are common default choices and we empirically find that they are not very sensitive to the final performance. The Dense Pivot Co-training does not include heavy engineering, as it can be simply implemented by inserting a dense training step and a distillation step every 2 mini-batches, and does not include any hyperparameter tuning.
>
>
>
> >Q10:Moreover, only by training IMP, I could potentially obtain multiple sparse networks...
>
> A10: As we have shown in our ablation study in Section 4.2, every part of Sparse Cocktail has an important contribution to the final performance. In essence, sparse co-training with IMP and weight-rewinding is necessary for a normal performance and every other part contributes to the performance boosts.
> We do not think one can transfer from unstructured sparsity to N:M sparsity during IMP following the mentioned approach in Q10.
> Notably, we do not increase the training time or iterations compared to vanilla IMP which solely trains unstructured sparsity, as Dense Pivot co-training utilizes the ***graident alignment*** effect as in the AST paper[2] using the ***alternative sparse training***. For example, in the original LTH paper, they used 182 epochs at each IMP iteration while in Sparse Cocktail we use 150 epochs with the same mini-batch size on the CIFAR10 dataset. Compared to other sparse co-training methods that can co-train multiple sparsity ratios (i.e. Mutual Net and AST), we also find they require a nearly full training schedule of 1500 epochs when co-training only 10 subnetworks.
>
> Hope our response can address your concerns!
>
> [5] The Lottery Ticket Hypothesis: Finding Sparse, Trainable Neural Networks, Jonathan Frankle, et al 2019.

---

### Meta-Review · Area_Chair_Cend · 2023-12-15

**Metareview:**

This work approaches the problem of learning sparse networks that are often designed for resource constrained systems. Instead of only learning one system, to make the same dense network compatible across many potential hardware constraints, multiple sparse networks with different sparsity constraints are trained simultaneously. Key to the approach is the use of processes to share parameters across these networks while maintaining performance across networks. The authors further apply their approach to a number of datasets.

The reviewers appreciated the problem formulation and overall results presented in the paper. While some more minor points were raised, the main point that was deliberated and was raised in different ways across reviewers was the potential novelty of the approach. Essentially, are the steps and combination novel, or is this a more straightforward application of existing techniques. While this was a subject of discussion during that period of the review, it seems that more work to clarify the novelty in the paper might be beneficial to convincing the reviewers more uniformly of the work's novelty. Therefore I do not recommend this paper for acceptance.

**Justification For Why Not Higher Score:**

The primary reason is the lack of confidence following the discussion period of the novelty of the approach.

**Justification For Why Not Lower Score:**

N/A

---

### Decision · Program_Chairs · 2024-01-16

Reject